# Dosage-Dependent Gynoecium Development and Gene Expression in *Brassica napus-Orychophragmus violaceus* Addition Lines

**DOI:** 10.3390/plants10091766

**Published:** 2021-08-25

**Authors:** Bowei Cai, Tai Wang, Wenqin Fu, Arrashid Harun, Xianhong Ge, Zaiyun Li

**Affiliations:** National Key Laboratory of Crop Genetic Improvement, National Center of Oil Crop Improvement (Wuhan), College of Plant Science and Technology, Huazhong Agricultural University, Wuhan 430070, China; caibowei_92@126.com (B.C.); 15296783715@163.com (T.W.); fuwenqinby@gmail.com (W.F.); harun_07bge@yahoo.com (A.H.); lizaiyun@mail.hzau.edu.cn (Z.L.)

**Keywords:** *Brassica napus*, gynoecium, monosomic alien addition line, female sterility, transcriptome analysis

## Abstract

Distant hybridization usually leads to female sterility of the hybrid but the mechanism behind this is poorly understood. Complete pistil abortion but normal male fertility was shown by one *Brassica napus-Orychophragmus violaceus* monosomic alien addition line (MA, AACC + 1 I_O_, 2n = 39) produced previously. To study the effect of a single *O. violaceus* chromosome addition on pistil development in different genetic backgrounds, hybrids between the MA and *B. carinata* (BBCC), *B. juncea* (AABB), and two synthetic hexaploids (AABBCC) were firstly produced in this study which show complete female sterility. A microspore culture was further performed to produce the haploid monosomic alien addition line (HMA, AC + 1 I_O_, 2n = 20) and disomic addition line (DA, AACC + 2 I_O_, 2n = 40) together with haploid (H, AC, 2n = 19) and double haploid (DH, AACC, 2n = 38) plants of *B. napus* from MA to investigate the dosage effect of the alien *O. violaceus* chromosome on pistil development and gene expression. Compared to MA, the development of the pistils of DA and HMA was completely or partially recovered, in which the pistils could swell and elongate to a normal shape after open pollination, although no seeds were produced. Comparative RNA-seq analyses revealed that the numbers of the differentially expressed genes (DEGs) were significantly different, dosage-dependent, and consistent with the phenotypic difference in pairwise comparisons of HMA vs. H, DA vs. DH, MA vs. DH, MA vs. DA, and MA vs. HMA. The gene ontology (GO) enrichment analysis of DEGs showed that a number of genes involved in the development of the gynoecium, embryo sac, ovule, and integuments. Particularly, several common DEGs for pistil development shared in HMA vs. H and DA vs. DH showed functions in genotoxic stress response, auxin transport, and signaling and adaxial/abaxial axis specification. The results provided updated information for the molecular mechanisms behind the gynoecium development of *B. napus* responding to the dosage of alien *O. violaceus* chromosomes.

## 1. Introduction

The development of a pistil begins with one or more fused carpels which have evolved from leaves [1]. In many species of Brassicaceae, including the model plant *Arabidopsis thaliana*, gynoecium consists of two fused carpels [2,3]. At the margin where the two carpels fuse, the carpel margin meristem (CMM) is formed, and the placenta, ovule, septum, and transmitting tract are then produced there [4]. The ovule develops from the ovule primordium cells and forms a finger-like structure at the early development stage which then differentiates into the funiculus, the chalaza, and the nucellus [5]. In the tip of the nucellus, the megaspore mother cell is differentiated from a hypodermal cell and undergoes meiosis followed by three rounds of mitosis to form a mature embryo sac comprised of three antipodal cells, two medial polar nuclei, one egg cell, and two synergid cells [6]. Pistil development is a complex process that requires several tissues to acquire specific identities in specific places. This process may be realized through the regulatory network formed by genes and hormones, including Auxins, cytokinins (CKs), gibberellins (GAs), and brassinosteroids (BRs) [7,8]. The genes can affect hormonal pathways at different levels, including biosynthesis, transport, and signaling, and in turn, the hormones can influence the transcriptional regulation of the genes.

Postzygotic reproductive isolation, such as hybrid sterility, hybrid necrosis/weakness, and hybrid lethality, are widely observed in crop plants, causing the reduced fitness of hybrids between divergent varietal groups. These barriers are unfavorable traits for the utilization of heterosis and are major obstacles to genetic crop improvement. Hybrid sterility represents the most common form of a postzygotic reproductive barrier in plants, and a well-characterized example is the hybrid sterility between two subspecies of Asian cultivated rice (*Oryza sativa* L.) [9]. According to the development stage, female sterility can be divided into three types: one is that the pistil does not differentiate at all or has an incomplete pistil; the second is that the pistil lacks a normal embryo sac and abortive ovule; the third is that the pistil has a normal embryo sac but the egg cell fails to develop into an embryo after pollination.

With the rapid development of sequencing technology, the research on genomes and transcriptomes becomes easier and more feasible because next-generation sequencing methods have dramatically increased the sequencing efficiency and reduced sequencing cost [10]. Transcriptome analysis is widely used to study the overall gene expression level of specific tissues, which is conducive to the overall analysis of the gene regulatory network of a certain character [11,12,13]. The analysis of pistil abortion by transcriptome has been reported in many species, including *A. thaliana* [14], *Oryza sativa* [15], *Prunus mume* [16], *Brassica rapa* [17], *B. napus* [18,19], *Olea europaea* [20], *Citrus reticulate* [21], and *Hieracium praealtum* [22]. All these studies show that numerous genes are involved in the regulation of pistil development, and the formation of the embryo sac and the development of the integument are interdependent, or partly controlled, by a common pathway, which indicates the complexity of pistil development regulation [23,24,25].

Rapeseed oil (*B. napus*, 2n = 38, AACC), an economically important oilseed crop, is widely cultivated around the world [26,27]. Due to the value of hybrid seeds in agriculture, a great deal of research has focused on male sterility in rapeseed but very few studies on pistil development have been reported in *B. napus* [28,29]. We previously reported a female sterile plant of the *B. napus*-*O. violaceus* monosomic alien addition line (MA) [30]. The results showed that the development of the female gametophyte was arrested after the tetrad stage and before megagametogenesis initiation [18]. Here, in order to study the effect of a single *O. violaceus* chromosome addition on pistil development in different genetic backgrounds, hybrids between MA and other Brassica species were produced and phenotypically observed. Meanwhile, a haploid monosomic alien addition line and disomic addition line were produced from the MA by microspore culture to investigate the dosage effect of the alien *O. violaceus* chromosomes on pistil development and gene expression changes in *Brassica napus* by phenotypic and comparative transcriptome analyses.

## 2. Materials and Methods

### 2.1. Plant Materials

The female sterile *B. napus*-*O. violaceus* monosomic alien addition lines (MA) contained the whole chromosome complement of *B. napus* (2n = 38, AACC) and one chromosome from *O. violaceus* (2n = 24, OO)(AACC + 1 I_O_, 2n = 39), which was previously developed after the somatic hybrid of these two species (2n = 62, AACCOO) was successively backcrossed by the parental *B. napus* cultivar “Huashuang 3” (H3) [30,31]. We previously reported the female sterile disomic addition line with one pair of the *O. violaceus* chromosomes (S1) [18]. However, the line was confirmed later as a double monosomic addition with two different *O. violaceus* chromosomes (data not shown). Furthermore, because the single chromosome addition was complete in the female sterile in a different genome background, disomic addition could not be obtained by selfing.

Hybrids between MA and two synthetic *Brassica* hexaploids (AABBCC, 2n = 54) [32,33] as well as *Brassica juncea* (AABB, 2n = 36) (accession number, GJ19) and *B. carinata* (accession number G0-7), were produced by hand pollination.

The female sterile monosomic additions (MA) were used for microspore culture and a total of 210 plants were produced, including four genotypic plants, namely haploid (H) and doubled haploid (DH) of *B. napus*, disomic additions (DA), and haploid monosomic additions (HMA). The detailed operation method of microspore culture referred to was Liu et al. (2003) [34]. Those plants with the *O. violaceus* chromosomes were identified by the SSR markers designed from its full-length transcriptome, and then confirmed by cytological observation. The expected 146 bp product of *O. violaceus* was amplified using specific SSR primers (forward primer ACGTAGCTTCCTCTCACTCTCCT and reverse primer TCAAATAATCAAGAACCGGTGAG). All the plants were planted in the experimental field of Huazhong Agricultural University in Wuhan, China.

### 2.2. Cytological Observation and GISH Analysis

Young ovaries for chromosome staining and counting and inflorescences for meiotic study were collected and fixed in a mixture of ethanol: acetic acid (3:1) for 24 h and then transferred to 70% ethanol and refrigerated until use. The determination of chromosome numbers and meiotic observations were made according to the method of Li et al. (1995) [35].

Slide preparations of chromosomes for GISH mainly followed the procedures of Zhong et al. (1996) [36] and Ge et al. (2007) [37]. In situ hybridization was carried out according to the protocols of Leitch et al. (1994) [38]. Hybridization signals of the *O. violaceus* probe were detected using Cy3-labeled streptavidin (Sigma, St. Louis, MO, USA) and chromosomes were counterstained with 0.2% 4′-6-diamidino-2-phenylindole (DAPI) solution (Roche, Basel, Switzerland), mounted in antifade solution (Vectashield), and examined under a Zeiss fluorescent microscope (Axio Scope A1, Munich, Germany) equipped with a CCD camera. Images were processed using Adobe Photoshop (8.0) to adjust contrast and brightness.

### 2.3. DNA Extraction and Probe Labelling

Total genomic DNA was extracted and purified from young leaves according to Dellaporta et al. (1983) [39]. The DNA from *O. violaceus* was labeled with Bio-11-dUTP by random priming using the Bio-Prime DNA Labeling System kit (Invitrogen, Life Technologies, Waltham, MA, USA) and used as a probe. The DNA of two hexaploids and one *B. juncea* was sheared by boiling for 15 min and used as a block.

### 2.4. RNA Extraction, cDNA Libraries Preparation, and Sequencing

According to Fu et al. (2014) [18], gynoeciums within flower buds from 1.5 to 3 mm in length were collected and immediately immersed in liquid nitrogen and then kept in a refrigerator at −80 °C until use. Total RNA was extracted using Trizol (Invitrogen, Burlington, ON, Canada) according to the manufacturer’s protocol. The quality and integrity of all RNA samples were assessed with a 2100 Bioanalyzer (Agilent Technologies, Santa Clara, CA, USA) and by electrophoresis on 1.5% agarose gels. For the cDNA library construction, 1.5 μg of total RNA per biological replicate was prepared according to the TruSeq RNA Sample Prep v2 protocol (Illumina). Subsequently, 15 libraries (three biological duplicates for each genotypic plant, MA, DH, H, HMA, and DA) were sequenced on an Illumina HiSeqTM 3000 platform (Illumina), to generate 150-bp paired-end reads. In order to obtain the genome sequence of the *O. violaceus*, full-length transcriptome sequencing was performed. High-quality RNA was extracted from six types of *O. violaceus* tissues (roots, stems, leaves, seeds, siliques, and inflorescences). Total RNA was synthesized to the first-strand cDNA using a Clontech SMARTer PCR cDNA Synthesis Kit. After PCR Optimization, a large-scale PCR was performed to synthesize second-strand cDNA for BluePippin^TM^ size selection. The ranges of size selection are 1~2 kb, 2~3 kb, 3~6 kb, 5~10 kb. After another large-scale PCR (if the transcripts were more than 3~6 kb, an optional Second BluePippin^TM^ Size Selection was needed) the DNA was ready for SMRTbell Template Preparation and sequencing. The full-length transcriptome sequencing was carried out by using a Pacific Bioscience RS II single molecular sequencing platform in BGI Co., Ltd., Beijing, China.

### 2.5. Full-Length Transcriptome Analysis

Qualified sequencing data produced by Pacific Biosciences RS II were processed following SMRT analysis through Reads of Insert, Classify, and Cluster, to obtain consensus full-length isoforms. Reads of Insert are used to process reads from the insert sequence of single molecules and estimate the length of the insert sequence loaded into an SMRT Cell, and then generate reads from the insert sequence of single molecules. In the filtering parameter, the minimum full pass is 0, and the minimum accuracy is 0.75.

Classify generates Reads of Insert from SMRT Cell cDNA molecules, removes cDNA primers and polyA sequences from reads, and then classifies Reads of Insert into full-length or non- full-length, chimeric or non- chimeric reads. The minimum sequence length is 300 bp, the minimum phmmer score to detect a primer in a read is 10. A Cluster module predicts de novo consensus isoforms from classified Reads of Insert using the ICE (Iterative Clustering and Error Correction) algorithm, then polishes predicted consensus isoforms using Quiver and classifies the polished isoforms into high QV or low QV isoforms based on user-specified criteria. The default minimum Quiver accuracy needed to classify an isoform as high quality is 0.99 to libraries below 3 kb, for libraries between 3~6 kb, the minimum Quiver accuracy is set to 0.98, for libraries between 5~10 kb, the minimum Quiver accuracy is set to 0.95. Because no reference genome of *O. violaceus* is available, we used the cd-hit-est to remove the redundancy based on sequence similarity. Because redundant transcripts are highly similar, the parameters as follows were used: cd-hit-est -i <input> -o <output> -c 0.98 -T 6 -G 0 -aL 0.90 -AL 100 -aS 0.98 -AS 30.

### 2.6. Reads Mapping and Differentially Expressed Genes (DEGs) Analysis

The *B. napus* genome(ZS11) [40], adding all CDS predicted from the full-length transcriptome of *O. violaceus*, was used as the reference genome. Trimmomatic version 0.33 [41] was used to remove adapters and low-quality reads in raw data. Clean reads were then aligned to the reference genome using HISAT version 0.1.6 [42]. Those unique mapped reads were used for the following gene expression analysis.

The R-package DESeq2 [43] was used to identify differentially expressed genes (DEGs) between five genotypic plants. Before identifying the DEGs, we used the R-packet corrplot to calculate the correlation among the three biological repeats of each sample to determine whether it was suitable for the next analysis. Then, the false discovery rate of q < 0.05 and the absolute value of log2Ratio ≥ 1 were taken as the threshold to judge gene expression difference. Due to the influence of plant ploidy and biological significance, we only calculated the DEGs of comparisons under the same ploidy [44]. For gene ontology (GO) annotation, a total of 101,040 genes from *B. napus* and 68,237 predictive genes from *O. violaceus*, were compared to the Swissport database. Next, the GO enrichment analysis of the DEGs was performed by TBtools version 0.66831 [45], with an adjusted *p*-value under 0.05 as a cutoff, to determine significantly enriched GO terms. The Kyoto Encyclopedia of Genes and Genomes (KEGG) [46] was used to explore the biological pathways the DEGs associated with. All genes were submitted to the KEGG Automatic Annotation Server (https://www.genome.jp/tools/kaas/) for the annotation documents of pathways in which genes were involved. KEGG enrichment analysis of DEGs was also performed by TBtools version 0.66831 [45] with *p*-value < 0.05 considered as a significantly enriched pathway.

### 2.7. Quantitative Real-Time PCR (qRT-PCR) Analysis

Complementary DNA (cDNA) was synthesized using a RevertAid First Strand cDNA Synthesis Kit (Thermo Scientific, Waltham, MA, USA) according to the manufacturer’s protocol. The Primer5 software was used to design the specific primers used for qRT-PCR (Appendix A). The Actin gene *BnENTH* was used as an internal reference control to normalize mRNA, and the 2^−ΔΔCT^ method was used to quantify the mRNA [47]. SYBR Green-based qRT-PCR was carried out using the Bio-Rad IQ5 real-time PCR detection system (Bio-Rad, Hercules, CA, USA) with the Luna Universal qPCR Master Mix (New England Biolabs, Ipswich, MA, USA). The PCR amplification reactions were performed in 96-well plates as the following cycling program: initial activation at 95 °C for 1 min, followed by 45 cycles of 95 °C for 15 s, 59 °C for 30 s, and 68 °C for 15 s. This procedure was followed by melting curve analysis from 60 to 95 °C to check the specificity of the product. All reactions were performed with three technical replicates and three biological replicates used to present the result for normalization, respectively.

## 3. Results

### 3.1. Complete Female Sterility in Different Brassica Species Caused by Single O. violaceus Chromosome

In order to explore the effect of the alien *O. violaceus* chromosome on gynoecia development in different genomic backgrounds, besides *B. napus*, hybrids between the female sterile MA and *B. carinata* (BBCC), *B. juncea* (AABB), and synthetic hexaploid 1 (AABBCC) derived from *B. carinata* and *B. rapa* (AA) [32], and hexaploid 2 from the sequential hybridization of three *Brassica* diploids [33], were produced. As a result, 36–44% of the hybrids were completely female sterile with short and abortive gynoecia as shown by MA (Figure 1A–F). Genomic in situ hybridization (GISH) analysis showed that all hybrids with female sterility had one *O. violaceus* chromosome in their pollen mother cells (Figure 1G–I). Therefore, the single *O. violaceus* chromosome was effective for female sterility in different *Brassica* species.

### 3.2. Dosage Effect of Alien Chromosome on Gynoecium Development

To obtain more addition lines, with single and two copies of the alien *O. violaceus* chromosomes in haploidy and diploidy backgrounds of *B. napus*, for studying the dosage effect on gynoecium development, we carried out a microspore culture of MA, doubled the chromosome number of the plantlets with colchicine treatment, and produced 210 plants. By PCR detection with the *O. violaceus* chromosome-specific primers, 178 (84.76%) plants were identified as harboring the *O. violaceus* chromosome, while the remaining 32 (15.24%) plants did not (Figure 2A). The ratio of plants with and without *O. violaceus* chromosomes was 5.56:1. The existence of the *O. violaceus* chromosome was further confirmed by cytological observation because it was relatively larger and darkly stained by modified carbol fuchsin (Figure 2B–D). Due to the influence of the planting environment, 162 of the 210 plants survived and were further identified by cytological observation and phenotype at the flowering stage. Among 162 plants, 10 (6.17%) disomic additions (DA) (AACC + 2 I_O_), and 125 (77.16%) haploid monosomic addition (HMA) plants (AC + 1 I_O_), were identified. In addition, we found 3 (1.85%) double haploid (DH) plants (AACC) and 24 (14.81%) haploid (H) plants (AC).

The flowers of the haploid plants of *B. napus* and HMA are small and have wizened anthers while all DH, DA, and MA plants in the *B. napus* diploid background showed larger flowers (Figure 3A) and normal anthers with abundant stainable pollen grains (90–95% stainability). Obvious stigma could be observed downward in the center of the flowers of all H, HMA, and DH plants but none in the MA and DA plants (Figure 3A). The size and shape of the gynoecia of H plants were normal as were those of the DH plants but the gynoecia of MA grew slowly and were much shorter from the early development stage (flower buds ~2 mm) to flowering as described previously [18] (Figure 3B). However, although the gynoecia of HMA and DA were shorter than those of DH and H plants they were longer than those of MA (Figure 3B). After the flowers opened, all the pistils of HMA and most of the pistils of DA showed an indication of swelling and elongation as did those of H and DH, but all the pistils of MA did not and fell off quickly (Figure 3C). However, HMA and DA produced no seeds after open pollination or by hand pollination (Figure 3D).

### 3.3. RNA-seq Analysis of O. violaceus and Additions

We firstly performed a full-length transcript analysis of *O. violaceus*. A total of 1,087,641,817 bp raw data was obtained by sequencing, including 349,123 reads, among which 134,706 were from the 1–2 k library, 125,147 from the 2–3 k library, and 89,270 from the 3–6 k library. After removing the low-quality sequences from the reads and removing the redundancy from the combined library, 68,237 isoforms with a total length of 148,152,103 bp and an average length of 2171 bp and N50 of 2694 bp were obtained. A total of 66,334 CDS sequences (unigenes) were predicted through functional annotations, with an average length of 790 bp (Appendix A).

In total, 15 RNA libraries were subjected to paired-end RNA sequencing, and 354.3 million clean reads were obtained with an average of 23.6 million reads (3.5 Gb) in each sample (Appendix A). The genome of *B. napus* (ZS11) [40] and de novo assembly *O. violaceus* CDS were integrated and used as the reference genome. The mapping rates of all samples were above 85% and approximately 81.1% of mapped reads were uniquely matched (Appendix A). The gene expression correlations between each pair of biological replicates were high with all Pearson correlation coefficients (R) >0.89 (Appendix A). These results indicated that the sequencing data of each biological replicate were of high quality. Those uniquely mapped reads were used for further analysis.

### 3.4. Gene Expression Changes in Additions Induced by the O. violaceus Chromosome

In order to reveal the potential mechanism of the abnormal development of pistils in additions, DEGs in pairwise comparisons of different combinations were identified (Figure 4). Generally, 2054, 4717, 8829, 8010, and 6715 DEGs were found in the comparisons of HMA vs. H, DA vs. DH, MA vs. DH, MA vs. DA, and MA vs. HMA, of which, 47.2% (969/2054), 25.8% (1218/4717), 8.5% (746/8829), 5.7% (455/8010), and 6.2% (413/6715) of genes were from *O. violaceus*, respectively. qRT-PCR analysis of twelve randomly selected DEGs presented very similar expressions in each sample as revealed by RNA-seq (Figure 5). GO enrichment analysis indicated that these DEGs in each comparison were mainly involved in DNA and RNA binding, motor activity, transcription regulator activity, transporter activity, and structural molecule activity (Appendix A). Meanwhile, these DEGs were mainly involved in the intracellular and extracellular regions, mitochondrion, nucleolus, nucleus, ribosome, and cell wall. Interestingly, several biological processes which were tightly regulated to pistil development were significantly enriched in DA vs. DH, MA vs. DA, and MA vs. DH, for example, embryo development, post-embryonic development, and fruit ripening (Appendix A).

### 3.5. DEGs Related to the Pistil Development

In order to further identify the genes related to pistil development, common DEGs were screened between HMA vs. H and DA vs. DH. These DEGs were only induced by the addition of one and two copies of chromosomes from *O. violaceus* under haploid and diploid backgrounds, respectively, and showed a relatively low number. As a result, 1086 common genes were identified, including 789 genes of *O. violaceus* and 297 genes of *B. napus* (Figure 6A). GO analysis found that several biological processes closely related to pistil development were significantly enriched, such as embryo development and post-embryonic development (Figure 6B). Similarly, 4784 common DEGs were screened between MA vs. HMA and MA vs. DA, including 296 genes of *O. violaceus* and 4488 genes of *B. napus* (Figure 6C). These DEGs were induced by the difference of the genome ploidy of *B. napus* and the number of additional *O. violaceus* chromosomes. However, GO analysis did not find any aspect of the biological process related to pistil development (Figure 6D).

We further investigated the common DEGs between DA vs. DH and HMA vs. H which are involved in embryo development and post-embryonic development. This revealed seventeen genes from *B. napus* (Appendix A) and seventy genes from *O. violaceus*. According to their expression patterns, those seventeen genes of *B. napus* were classified into three groups (Figure 7A; Appendix A). The first group included four genes that were lowly expressed in H and DH but significantly highly expressed in HMA, MA, and DA, including *MIRO1*, *FIS1A*, *BOP1* genes, and one *CRF6* gene. The second group included four *INOs*, *AGL11*, and *MEE3*, which were highly expressed in H and DH but silenced or weakly expressed in HMA, MA, and DA. The third group included seven genes that were silenced or weakly expressed in H and DH but significantly highly expressed in HMA, DA, and MA lines, including *COG7*, *OVA6*, *PIN1*, *PPRT3*, *ACC1*, *NAC085*, and another *CRF6*. Seventy DEGs from *O. violaceus* were only expressed in MA, DA, and HMA, which had an additional *O. violaceus* chromosome but were silenced in H and DH (Figure 7B; Appendix A).

### 3.6. Down-Regulation Genes for Brassinosteroid Biosynthesis in MA

In a previous study, Brassinosteroid (BR) biosynthesis and the metabolic process were enriched in DEGs between MA and DH, and twelve unigenes, which encoded seven kinds of proteins (LUP2, CAS1, SMO1, FACKEL, SMO2, DWF5, and DWF1), were down-regulated in MA [18]. The expression of genes for these proteins was also checked in each line. As a result, among forty-six genes encoding the above proteins, six genes were differentially expressed in MA vs. DH and MA vs. DA, and nine genes were differentially expressed in MA vs. HMA. (Appendix A). Except for two genes for *SMO1-1*, which were up-regulated in MA in comparison with DH, DA, and HMA, the expressions of all other genes were down-regulated in MA.

## 4. Discussion

In this study, we revisited the female sterile *B. napus*-*O. violaceus* addition line together with disomic addition and haploid monosomic addition developed by microspore culture as well as hybrids between MA and other Brassica species. This revealed very different gynoecium development and growth process which indicated these lines were valuable for elucidating the genetic mechanisms behind the gynoecium development and female sterility in wide hybrids involving *Brassica* crops and other crucifers [18].

### 4.1. Genetic Effects of Alien Chromosomes in Different Brassica Species and Dosage Effect of Alien Chromosome on Gynoecium Development

It was shown that the addition of one chromosome of *O. violaceus* could cause complete female sterility in *B. napus* as well as in the nullisomic line with one chromosome pair lost (2n = 36) of *B. napus* [18]. Anatomical observation showed that the development of ovules failed at a very early stage, in which the inner and outer integument did not develop and the megasporogenesis was blocked after the tetrad stage [18]. Here, it was found that this chromosome also caused complete female sterility in the hybrids with one *O. violaceus* chromosome between MA and *B. juncea* (AABB), MA and *B. carinata* (BBCC), as well as between MA and the synthetic allohexaploid (AABBCC). These results suggest that the genes on this alien chromosome for female sterility were still active in the different genetic backgrounds of the *Brassica* species.

In order to further investigate the effect of the single *O. violaceus* on pistil development under a genome background with different ploidy, microspore culture was then performed on MA to obtain disomic and haploid monosomic addition plants. Interestingly, in the DA and HMA, the development of the pistils was partly or completely recovered, and the pistils did not fall off after flowering despite their shorter length than those of the H and DH lines. Particularly, the pistils of DA and HMA could swell and elongate after open pollination. These results indicate that the pistil growth and elongation in different additions were affected not only by the alien chromosome itself but also by the ratio between the ploidy of the *B. napus* genome and the chromosome number of *O. violaceus*. Generally, three weeks after pollination, obvious swelling in the pod of *B. napus* could be seen with the young seeds inside. Swelling in the pods of H plants (Figure 3D) might have resulted from the growth of the seeds derived from the unreduced gametes. However, in the young pods of DA and HMA, no such swelling was observed (Figure 3D). It was speculated that the development of the ovule/female gametophyte (FG), which resulted from the failure of the integument and megagametogenesis or one of them, was still a failure in DA and HMA. Of course, other pre- and post-fertilization disorders may lead to the failure of seed production, such as poorly developed stigmas and abortion of the zygote, which need to be confirmed in the future.

Interestingly, the rate of the microspore-derived plants containing the *O. violaceus* chromosomes was much higher than the theoretical expectation ratio of 1:1. This result suggests the preferential retention of the *O. violaceus* chromosome happened during the meiotic division of the MA. In a *wheat-Aegilops* MAAL, it was found that gametes containing the *Aegilops* chromosome could be preferentially transmitted, resulting in the infertility of gametes that did not contain the *Aegilops* chromosome. This chromosome is called the gametocidal chromosome [48]. However, because the pollen fertility of *B. napus*-*O. violaceus* MAAL was not significantly lower than that of *B. napus* [18], it was unlikely that the *O. violaceus* chromosome played such a gametocidal role. The higher percentages of DA vs. DH plants (6.17 vs. 1.85) and HMA vs. H plants (77.16 vs. 14.81) confirms that the alien chromosome from *O. violaceus* favors plant regeneration from microspore cultures. This could be another reason for a higher percentage of plants containing *O. violaceous* chromosomes. On the contrary, very low percentages of plants having alien chromosomes were identified in the microspore-derived plants of the *B. napus*-*Crambe abyssinica* addition line [49] and *B. napus*-*Sinapis alba* addition line [50]. It is possible that *O. violaceus* itself has a strong plant regeneration ability during the microspore culture process, but there is no related result at present.

### 4.2. Global Gene Expression Changes upon the Alien Chromosome and Genome Ploidy

Due to the interaction of transcription networks, interspecific hybridization, and polyploidization led to widespread global gene expression changes. The changes were likely caused by *cis*-regulatory elements including gene expression enhancers and promoter sequences and trans-regulatory factors, such as transcription factors (TFs) [51]. Here, one or one pair of the alien chromosomes led up to 8083 genes with changed expression levels in *B. napus.* These transcriptional changes were caused by *trans*-regulatory factors carried by the additional *O. violaceus* chromosome. Intriguingly, it was found that gene expression changes from the chromosomal addition were very different in various genomic combinations. The two lowest numbers of DEGs were identified in HMA vs. H and DA vs. DH, indicating that dramatic gene expression changes of *B. napus* occurred when the rate between the number of the *O. violaceus* chromosomes and ploidy of the *B. napus* genome deviated from 1:1. This was consistent with the phenotypic difference where pistils were normal in H and DH but completely aborted in MA, and elongated and swelled in DA and HMA. Similarly, the numbers of DEGs from *O. violaceus* were also very different (1218 in DA vs. DH, 746 in MA vs. DH, and 969 in HMA vs. H). We think the differences may be caused by at least two reasons: (1) the different chromosome number of *O. violaceus* in DA, MA, and HMA, with 2, 1, and 1, respectively; (2) the different ploidy of the background genomes, haploidy (HMA), and diploidy (DA and MA). These observations argued for the gene balance hypothesis, which posits that dosage imbalances of genes encoding regulatory molecules disturb their stoichiometry within multi-protein complexes and disrupts cellular processes [52]. For example, in *Arabidopsis* aneuploids, triplicated chromosome 5 disrupted gene expression throughout the genome compared to normal diploids [53]. Moreover, the number of total DEGs (8829) identified here, in MA vs. DH, was significantly more than that in the previous study where 4540 differentially expressed unigenes were identified [18]. This might result from different materials and analysis methods used.

### 4.3. Gene Activation and Silencing Related to Female Sterility in Additions

Genes in pathways related to BR, auxin metabolism, and signaling, as well as adaxial/abaxial axis specification, were revealed previously and were most likely responsible for the abortive development of female organs [18]. Here, by discriminating those DEGs of *B. napus* from *O. violaceus*, 17 genes of *B. napus* related with ovules or FG development were identified in common DEGs between DA vs. DH and HMA vs. HA, which might play key roles in female sterility. Seven of these genes showed up-regulated expression in MA, DA, and HMA, including *AAC1*, *MIRO1*, *FISA1*, *BOP1*, *PPRT3*, and two *CRF6* genes. The mutants of these genes in *Arabidopsis* usually showed the arrest of ovule or embryo development, such as *AAC1* [54] and *MIRO1* [55]. However, the relationships of the up-regulated expression level in additions and the female sterile phenotype were unclear. Another gene, *AGL11*, was down-regulated in MA, DA, and HMA. *AGL11* encoded a MADS-box transcription factor expressed in the carpel and ovules and controlled the structure and mechanical properties of the seed coat [56]. The down-regulation of the gene might result from the arrest of ovule development here.

Four (*PIN1*, *NAC085*, *OVA6*, and *COG7*) and five genes (four *INOs* and *MEE3*) were activated and silenced, respectively, in the MA, DA, and HMA, which might play key roles in female sterility in additions. Auxin had wide-ranging effects on growth and development throughout the plant, including gynoecium and ovule morphogenesis [57,58,59]. Auxin could diffuse cell-to-cell actively by two types of transport, efflux and influx carriers. PIN1 was a known efflux carrier that played key roles in auxin signaling, in the absence of this auxin transporter, arrested FG was observed [60,61]. Recent studies indicate that auxin efflux played the core role in precisely controlling the spatiotemporal pattern of auxin distribution in the nucellus surrounding the FG. The auxin efflux carrier PIN1 transported maternal auxin into the nucellus while PIN3/PIN4/PIN7 further delivered auxin to degenerating nucellar cells and concurrently controlled FG central vacuole expansion [62]. In the H and DH at the 1 mm pistil development stage, *PIN1* was almost silenced but was highly expressed in HMA, MA, and DA, which might disturb the normal auxin signaling and led to the failure of FG development. NAC085 was a NAC-type transcription factor that was required for DNA damage and other types of stress, such as heat stress-induced G2 arrest [63]. The significantly high expression of *NAC085* in MA, DA, and HMA might be a result of genotoxic stress response to alien chromosome addition and led to the cell cycle arrest in ovule development. *OVA6* and *COG7* encoded a chloroplast and mitochondria localized prolyl-tRNA synthetase and a subunit of the conserved oligomeric Golgi (COG) complex, respectively. Both mutants could lead to embryo development arrest and a yellow embryo [64,65]. However, again, the relationships of the up-regulated expression level and female sterility were unclear here.

Four *INO* genes showed the same expression pattern which was silenced in plants with one or two copies of *O. violaceus* chromosomes. This indicated that there might be one common regulator which repressed the expression of four genes. *INO* encoded a YABBA family protein and was expressed at the outer integument initiation site of the ovule primordium and at the abaxial side of integument cells at all developmental stages, playing a central role in regulating outer integument development [66,67,68]. A recent study indicated that *INO* not only affects the ontogenesis of the embryo sac but also restricts cell number and nucellus volume during nucellus development, although the mechanism was not clear [69].

### 4.4. Dosage-Dependent Gene Expression for BR Biosynthesis and Pistil Growth

Brassinosteroids (BRs) played important roles in regulating plant reproductive processes. BR signaling or BR biosynthesis null mutants did not produce seeds under natural conditions and BRs regulated outer in-tegument growth partially via BZR1-mediated transcriptional regulation of INO [70]. Several unigenes, which encoded kinds of proteins for BR synthesis, were down-regulated in S1(MA in this study) [18], but GO and KEGG enrichment analysis (data not shown) did not identify BR signaling pathway genes in common DEGs between HA vs. HMA and DA vs. DH here. This might result from the fact that most of the genes involved in BR biosynthesis were only down-regulated in MA, in comparison with DA, DH, and HMA (Appendix A). These results also indicated the expressions of the genes for BR biosynthesis were dosage-dependent on the ratio of the number of *O. violaceus* chromosomes and the genome ploidy of *B. napus*. In consideration of the fact that the swelling and elongation of the pistil failed only in MA, it was thought that pistil swelling and elongation might depend largely on BR biosynthesis and signaling.

In conclusion, the phenomenon of female sterility in *B. napus* induced by one specific *O. violaceus* chromosome was revisited in several hybrids between MA and other Brassica species, as well as addition lines developed successfully with a microspore culture of MA. Phenotypic results showed that the growth of the pistils was recovered in DA and HMA which showed a dosage-dependent development. Comparative RNA-seq analysis in different addition lines revealed that the genes for the pathways related to auxin signaling and *NAC085*-mediated cell cycle arrest were most likely responsible for female sterility, while the pistil swelling and elongation might be regulated mainly by BR biosynthesis and signaling pathway genes.

## Figures and Tables

**Figure 1 plants-10-01766-f001:**
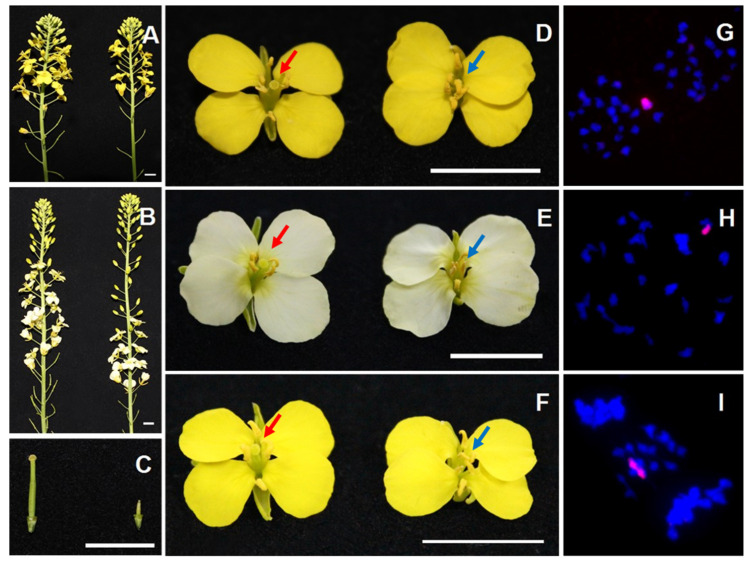
Morphology and cytology analysis of offspring of the hybridization between female sterility addition line and different plants containing B genome. (**A**,**B**) Inflorescence of offspring between MA and hexaploid 1 derived from *B. carinata* and *B. rapa* (AA) or hexaploid 2 from the sequential hybridization of three *Brassica* diploids, respectively. (**C**) Gynoecium of offspring between MA and hexaploid 1. (**D**–**F**) Flowers of offspring between MA and hexaploid 1 or hexaploid 2, or *B. juncea*, respectively. (**G**–**I**) The pollen mother cells of female sterile plants corresponding to (**D**–**F**), respectively. Red signals were from the labeled *O. violaceus* probe and blue color from DAPI staining. (**A**–**F**) Female fertile and sterile plants were on left and right, respectively. The red and blue arrows in the D-F indicate the normal pistils and invisible pistils, respectively. Bar = 1 cm.

**Figure 2 plants-10-01766-f002:**
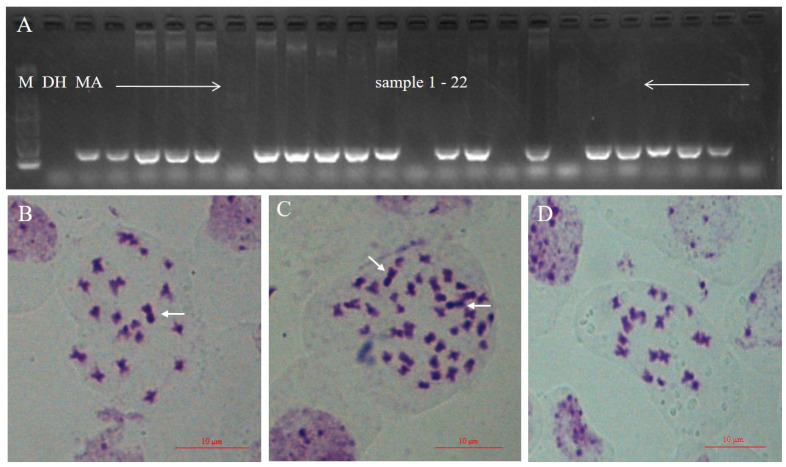
Identification of plants with *O. violaceus* chromosome by molecular marker analysis and cytological observation. (**A**) SSR marker detection of plants with and without *O. violaceus* chromosome. (**B**–**D**) One somatic cell at mitotic metaphase of HMA(AC + 1 I_O_, 2n = 20), DA(AACC + 2 I_O_, 2n = 40), and H(AC, 2n = 19), respectively. The white arrow points to the *O. violaceus* chromosome which is relatively larger and blacker than the others. Scale bars = 10 µm. M, maker; DH, doubled haploid of Huashuang3; MA, monosomic addition.

**Figure 3 plants-10-01766-f003:**
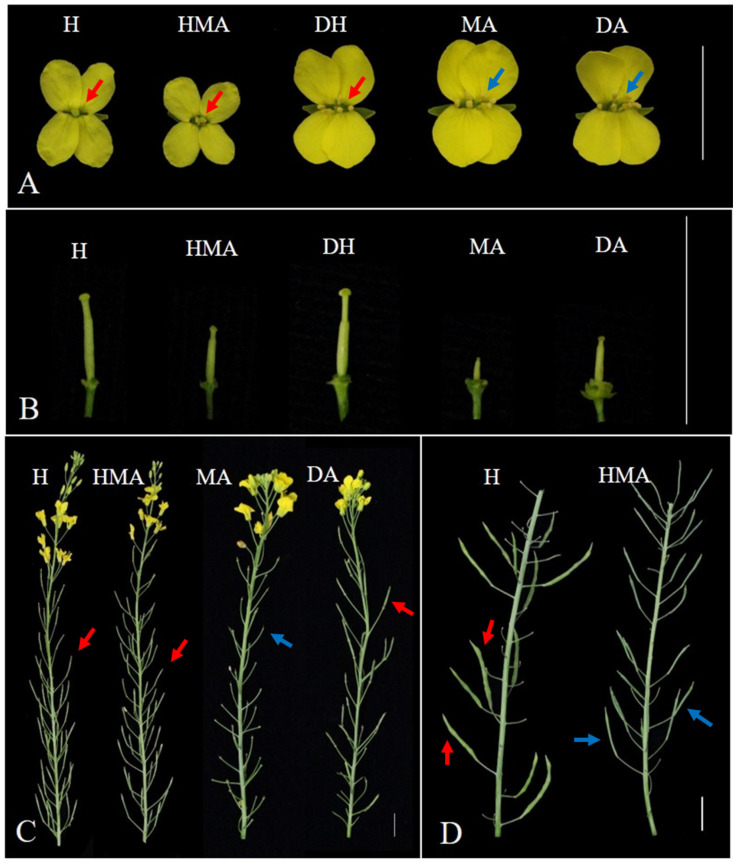
Phenotypic observation of flowers and pistils of H (AC), HMA (AC + 1 I_O_), DH (AACC), MA (AACC + 1 I_O_), and DA (AACC + 2 I_O_), respectively. (**A**,**B**) Flowers and pistils in flower buds at 10 mm stage of different lines. (**C**,**D**) Pistil/Pods at late flowering stage (**C**) and three weeks after pollination (**D**) of different lines. The red and blue arrows indicate the normal pistils and invisible pistils in A, the elongated pistils and pod stalk after pistils fell off in C, and swelling on silique and invisible swelling in D, respectively. Bar = 20 mm.

**Figure 4 plants-10-01766-f004:**
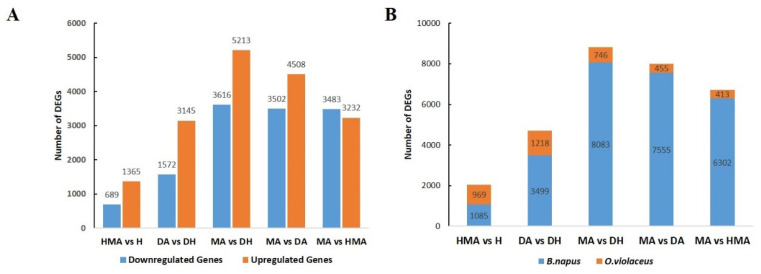
DEGs and their origin identified in HMA vs. H, DA vs. DH, MA vs. DA, and MA vs. HMA. (**A**) The number of up- and down-regulated genes in each comparison. (**B**) The origin (*B. napus* vs. *O. violaceus*) of DEGs in each comparison.

**Figure 5 plants-10-01766-f005:**
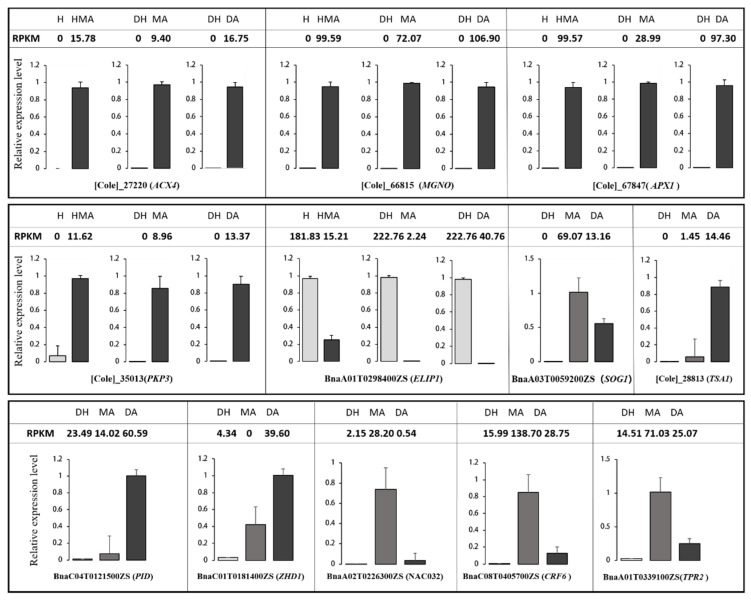
qRT-PCR confirmation of randomly selected DEGs. Expression values (reads per kb per million reads, RPKM) of 12 DEGs revealed by RNA-seq analysis are shown by the numbers on the top of each gene while the relative expression level revealed by qRT-PCR are represented by the columns. Columns and bars represent the means and standard error (*n* = 3), respectively. H (AC), HMA (AC + 1 I_O_), DH (AACC), MA (AACC + 1 I_O_), DA (AACC + 2 I_O_).

**Figure 6 plants-10-01766-f006:**
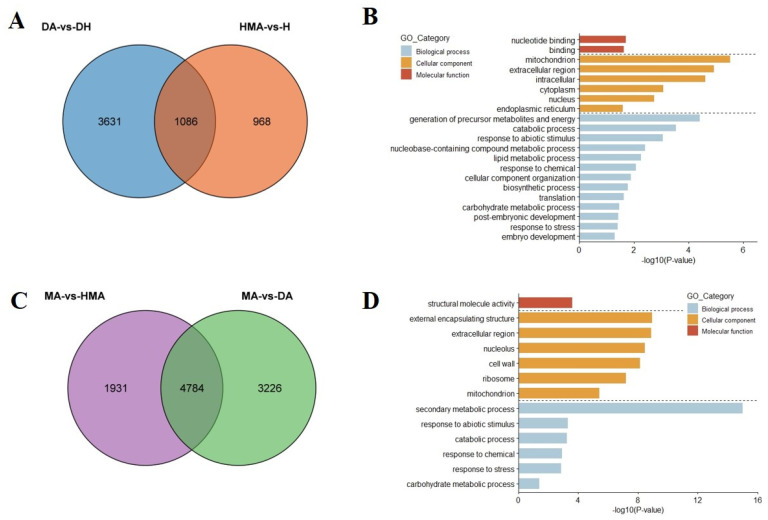
Identification and GO enrichment analysis of common DEGs in DA vs. DH, HMA vs. H, MA vs. HMA, and MA vs. DA. (**A**,**B**) Venn diagram for the DEGs in DA vs. DH and HMA vs. H (**A**) and GO enrichment analysis of common DEGs (**B**). (**C**,**D**) Venn diagram for the DEGs in MA vs. HMA and MA vs. DA (**C**) and GO enrichment analysis of common DEGs (**D**).

**Figure 7 plants-10-01766-f007:**
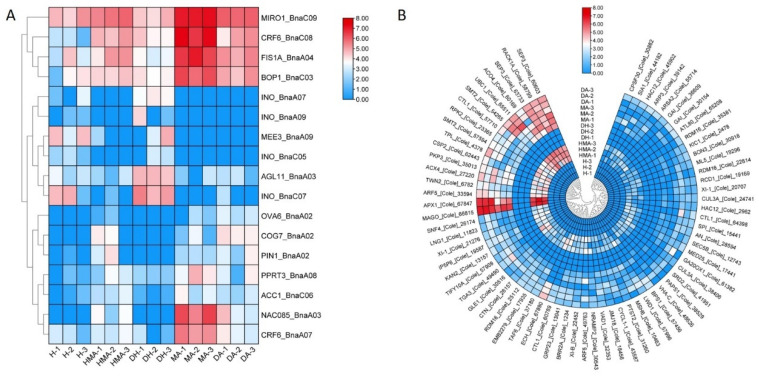
Clustering analysis of common DEGs between DA vs. DH and HMA vs. H enriched in GO terms involved in embryo development and post-embryonic development. (**A**) Common DEGs of *B. napus*. (**B**) Common DEGs of *O. violaceus*.

## Data Availability

Transcriptome data were deposited in NCBI under accession number: PRJNA637700.

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
