# Peer review of "Dosage-Dependent Gynoecium Development and Gene Expression in Brassica napus-Orychophragmus violaceus Addition Lines"

_plants, 2021, doi:10.3390/plants10091766_

Round 1

Author Response

Response to Reviewer 1 Comments  (You can see the attachment for the revision of the manuscript

Major questions:

Point 1: The DEGs between DA vs DH, MA vs DH, there was no Io chromosome in DH, but the numbers of DEGs of DA and MA were quietly different (1218 vs 746), in addition, HMA vs H, the DEGs were 969. Since there were no Io in DH and H, that means if there was expression of genes in Io, then that should be different, but why the numbers were so different (1218, 746, 969). These results should be discussed, were some silenced by the chromosome imbalance or some other reasons?

 Response 1: Good question. We think the difference may be caused at least by two reasons: (1) The chromosome number of O. violaceus is different in DA (2), MA (1) and HMA (1); (2) Ploidy of the background genomes are different, haploidy (HMA) and diploidy (DA, and MA). We have added several sentences in the discussion.  

Point 2:  In MM, the calculation methods for the relative expression between target gene and reference gene should be given, “ΔΔCt=……..”. It is more reliable if the author can provide the efficiency test of the reference control, because it must be hard to make cDNA concentrations consistent in different runs, but if the amplification efficiency is different at different concentrations, the results will affected. In addition, why did the authors select the 12 genes, randomly or any other reasons?

Response 2: Thanks. We have added the calculation in the MM part. In order to verify the reliability of the transcriptome data quality, we performed qPCR experiments on randomly selected DEGs.

Point 3: Page 8, paragraph 2: Did all (100%) the flowers of H, HMA and DH have stigma and none of MA, DA? The ratios should have been given in the results. For example, you might count 100 flowers, how many of them had perfect stigma?

Response 3: Yes, at the flowering stage obvious stigma could be observed downward in the center of the flowers of all H, HMA and DH plants but no one in MA and DA plants because the pistils of MA and DA are short. However, at present, we did not have relevant data.

Point 4: I suggest authors to discuss a little more about the phenomenon that DA and HMA produced good pistils but no seeds, because in canola industry, seeds are the good sweets. In theory, DA can produce a total set of chromosomes, why it cannot be producing seeds should be an quite interesting field to explore.

Response 4: Good suggestion! We have added several sentences in the paragraph. It was speculated that the developments of ovule/female gametophyte (FG) which resulted from the failure the integument and megagametogenesis or one of them is still failure in DA and HMA because no seed was produced. Of course, many other pre- and post fertilization disorders could lead to the failure of seed production, such as poorly developed stigmas and abortion of the zygote. More anatomical observations should be performed in future to answer the question.  

 Minor question:

  1. Italicize all the “B. napus

Response1:  We have revised it to B. napus.

  1. For all the numbers bigger than 1,000, add a comma every three digits, for example, 101040 should be 101,040, 68237 should be 68,237.

Response2: We have revised those numbers according to your advice.

  1. Use “Complementary DNA” at the beginning of a paragraph instead of cDNA.

Response3: We have revised it according to your advice.

  1. In the last paragraph of MM, you should note that the qPCR was sybr-green based.

Response4: We have revised it according to your advice.

  1. Figure 1: I like the comparisons between sterility and fertility flowers, but there is no purple stem as shown in the pictures??

Response5: Yes, there is little confused. We have changed the explanation. In fact, the purple colour is not obvious at flowering stage.

  1. Figure 2: change the order between “cytological observation” and “molecular marker analysis” in captions to be consistent to the pictures show. There is a double space on “A”, and embolden “B” and“D”.

Response6: We have revised it according to your advice.

  1. What are the O.violaceus chromosome specific primers? What was the conventional PCR product size?

Response7: Thank you so much for raising this question! We have added the sentence “The expected 146bp product of O. violaceus was amplified using specific SSR primers (forward primer ACGTAGCTTCCTCTCACTCTCCT and reverse primer TCAAATAATCAAGAACCGGTGAG)” in the text.

  1. 32 plants did not have O. violaceus, account for 15.2%, not 15.3%.

Response8: Thanks. We have change it to 15.24%.

  1. The ratios of different plants among 178 with O.violaceus were very confusing, how did the authors calculate the ratios? 10/178=5.62%, not 6.17%, even 10/210=4.76%, other ratios were also confusing.

Response9:  Due to the influence of the planting environment, 162 of the 210 plants survived finally and were further identified by cytological observation and phenotype at the flowering stage. So, all the ratios are calculated based on this total number of 162. We have made correspondingly revision in the text.

  1. Page 8, second paragraph, line 3: should be Figure 3A, not 2A.

Response10: Thanks. We have corrected it!

  1. 11. Page 8, second paragraph, line 12: did you do the significance test? If yes, provide p value, if not; do not use the word “significantly”.

Response11: We've removed the “significantly” here.

  1. Page 8, second paragraph, line 15: there was no DA but H on Figure 3D. Did H produce some seeds?

Response12: Yes, there is no DA in Figure 3D, because the siliques of DA at the later stage are the same as those in Figure 3C, so we did not put it in the figure 3D. Yes, H can produce a small number of seeds.

  1. Put Table 1 in supplementary.

Response13: We have put it in supplementary.

Response to Reviewer 2 Comments

Point 1: The introduction is too long. The marked fragment of the text is not needed, it is commonly known knowledge.

Response: We have deleted the paragraph.

Point 2:  The goal of the work should be better defined. At present form is mixed with results.

Response 2: Good suggestion. We have added one more sentence for the goal of the study.

Point 3: According to the manufacturer’s instructions (company name)

Response: Very good suggestion for modification! We had added corresponding information. The DNA from O. violaceus was labelled with Bio-11-dUTP by random priming using the Bio-Prime DNA Labeling System kit (Invitrogen, Life Technologies) and used as probe”.

Point 4: This antifade solution is Vectashield (Vector Laboratories)

Response 4: Thanks! We have made corrections in the text.

Point5: Figure 1 legend needs better formatting

Response: Thanks. We have made relevant changes.

Point6: That is part of methods. Mitotic polyploidization is not sufficiently described in Materials and Methods.

Response 6: Good suggestion, we have made a supplementary explanation in the Materials and Methods. The detailed operation method of microspore culture was referred to Liu et al.(2003).

Point7: The title of Figure 2 should be rewritten as follows 'Identification of plants with O. violaceus chromosome by molecular marker analysis and cytological observation.  Fist you show molecular marker then cytological.

Response: Thanks. We have changed it according to your advice.

Reviewer 2 Report

Overall, the manuscript is well-written. The results are clearly explained and discussed.  I have made some comments and suggestion in the pdf file, and hope they can help improving the current version. 

Author Response

Response to Reviewer 2 Comments (You can see the attachment for the revision of the manuscript

Point 1: The introduction is too long. The marked fragment of the text is not needed, it is commonly known knowledge.

Response1: We have deleted the paragraph.

Point 2:  The goal of the work should be better defined. At present form is mixed with results.

Response 2: Good suggestion. We have added one more sentence for the goal of the study.

Point 3: According to the manufacturer’s instructions (company name)

Response3: Very good suggestion for modification! We had added corresponding information. “The DNA from O. violaceus was labelled with Bio-11-dUTP by random priming using the Bio-Prime DNA Labeling System kit (Invitrogen, Life Technologies) and used as probe”.

Point 4: This antifade solution is Vectashield (Vector Laboratories)

Response 4: Thanks! We have made corrections in the text.

 Point5: Figure 1 legend needs better formatting

Response: Thanks. We have made relevant changes.

Point6: That is part of methods. Mitotic polyploidization is not sufficiently described in Materials and Methods.

Response 6: Good suggestion, we have made a supplementary explanation in the Materials and Methods. The detailed operation method of microspore culture was referred to Liu et al.(2003).

Point7: The title of Figure 2 should be rewritten as follows 'Identification of plants with O. violaceus chromosome by molecular marker analysis and cytological observation.  Fist you show molecular marker then cytological.

Response7: Thanks. We have changed it according to your advice.

Reviewer 3 Report

This manuscript can be accepted in present form.

Author Response

Thank you very much for your valuable comments! You can see the attachment for the revision of the manuscript!
